# Psychosocial Risk Factors and Burnout Among Teachers: Can Emotional Intelligence Make a Difference?

**DOI:** 10.3390/ijerph22091439

**Published:** 2025-09-16

**Authors:** Carla Barros, Carina Fernandes, Pilar Baylina

**Affiliations:** 1Faculty of Human and Social Sciences, University Fernando Pessoa, Praça de 9 de Abril 349, 4249-004 Porto, Portugal; carinafernandes@ufp.edu.pt; 2ESS, Polytechnic of Porto, Rua Dr. António Bernardino de Almeida, 400, 4200-072 Porto, Portugal; pilarbaylina@ess.ipp.pt

**Keywords:** psychosocial risk factors, burnout, emotional intelligence, teachers

## Abstract

Teaching is a complex profession that demands simultaneous cognitive and emotional efforts. The present study aims to determine whether teachers’ emotional intelligence moderates the relationship between psychosocial risk factors and burnout. A cross-sectional online survey was conducted among 215 secondary school teachers. Measurement instruments included the Burnout Assessment Tool (BAT-23) to assess burnout dimensions; the Health and Work Survey (INSAT) to evaluate psychosocial risk factors; and the Wong and Law Emotional Intelligence Scale (WLEIS-P) to assess emotional intelligence. A mediation/moderation analysis using the PROCESS macro was conducted to examine whether emotional intelligence mediates/moderates the relationship between psychosocial risk factors and burnout among teachers. The results show that psychosocial risk was a significant positive predictor of burnout (B = 0.313, *p* = 0.001), indicating that higher perceived risk was associated with higher burnout symptoms. Emotional intelligence did not significantly predict burnout on its own (B = 0.176, *p* = 0.364), and the interaction term (psychosocial risk × emotional intelligence) was not significant (B = 0.000, *p* = 0.995), suggesting that emotional intelligence does not moderate the relationship between psychosocial risks and burnout. These findings underscore a more holistic approach to address burnout, centered in intervention strategies that include a deeper analysis of organizational context determinants.

## 1. Introduction

Teaching is a complex profession that demands simultaneous cognitive and emotional efforts, requiring regular managing of professionals’ emotions as a key element in reaching educational objectives and fostering positive student outcomes [1].

Considering this complexity, several studies have highlighted the impact of burnout on the teaching profession. For example, a recent scoping review [2] reported prevalence rates ranging from as low as 2.81% [3] to as high as 70.9% [4]. Burnout is a multidimensional condition caused by a prolonged response to chronic emotional and interpersonal work-related stress, characterized by overwhelming exhaustion, interpersonal detachment, or cynicism towards the job, leading to a diminished sense of professional efficacy and personal fulfillment [5,6].

Burnout among teachers has been linked to a variety of factors, which include workplace-related factors such as years of teaching experience, class size, job satisfaction, the subject taught, conflicting beliefs, disintegration of the workplace, strained relationships with coworkers, loss of autonomy, insufficient compensation, absence of equity, high workload and time limits, resource shortages, fear of violence, student behavior issues, role ambiguity and conflict, limited opportunities for advancement, inadequate support, and low participation in decision-making [7,8,9,10,11,12]. In recent years, the challenges faced by teachers have intensified, particularly due to the additional pressures brought on by the COVID-19 pandemic [13]. Such work-related stressors are often pervasive. Teachers face heavy workloads, role ambiguity, large classes with diverse learning needs, limited organizational support, and frequent classroom management difficulties. Prolonged exposure to these adverse conditions erodes teachers’ mental and physical resources, eventually precipitating burnout syndrome [14,15,16].

Higher levels of occupational stress have been associated with increased rates of absenteeism, staff turnover, higher intentions to quit, lower teaching effectiveness and satisfaction, and numerous negative health effects, including fatigue, sleep disturbances, and hormonal imbalances [2]. These issues not only affect teachers’ health and personal lives but also impair their job performance and productivity, with indirect consequences for students, such as the quality of education they receive. Since ongoing burnout can evolve into mental health disorders such as anxiety and depression [5,15], this context underscores the importance of identifying and understanding the protective factors that contribute to mitigating the impact of burnout on teaching profession.

As mentioned above, one critical set of contributors to burnout are psychosocial risk factors in the work environment. The present study aims to systematically assess these risk factors, covering dimensions of the work environment (e.g., work pace, organizational climate, relational support, emotional demands) in the education sector to identify areas of psychosocial strain. Prior research has found that high levels of psychosocial risks at work tend to correlate with negative outcomes like stress, musculoskeletal disorders, and burnout symptoms [17,18]. These findings reinforce the conclusion that unfavorable psychosocial working conditions are risk factors for teacher burnout, warranting interventions at the organizational level.

At the same time, the present study aims to investigate the personal resources that might help teachers to cope with stress and resist burnout. Among these, emotional intelligence has emerged as a potentially important protective factor in the teaching profession [15,19,20]. Emotional intelligence broadly refers to the capacity to recognize, understand, and manage emotions in oneself and others and encompasses a range of self-perceived, emotion-related abilities that enable individuals to recognize, interpret, process, and use them [21,22]. Teachers with higher emotional intelligence are thought to better navigate the emotional demands of the classroom. For instance, they are better equipped to calmly resolve conflicts, sustain motivation, and recover from setbacks, which reduces their susceptibility to burnout.

A systematic review has demonstrated negative associations between emotional intelligence and burnout, suggesting that educators with higher emotional intelligence tend to report lower exhaustion and depersonalization [15,23]. Considering these results, emotional intelligence can help teachers reduce burnout and stay engaged in their work [24], likely by enabling more effective stress management, emotional regulation, and use of social support. Given these benefits, emotional intelligence is increasingly seen as valuable personal competency for teachers.

However, these findings are mostly based on cross-sectional and correlational studies, and to our knowledge, no studies have directly tested whether emotional intelligence mediates or moderates the relationship between psychosocial risk factors and burnout in teachers. This lack of evidence regarding its potential mechanisms provided the rationale for the present study: while correlations justified the expectation of a protective role of emotional intelligence, our work will empirically test its mediating and moderating role. In other words, it remains underexplored whether a teacher’s emotional intelligence can moderate (i.e., buffer or amplify) the impact of adverse psychosocial work conditions on their burnout levels. We may infer that personal resources might act as a buffer against organizational stressors and endorse protective factors for psychological well-being [15,25,26,27], but direct empirical investigations in educational settings are scarce.

To address this gap, the present study aims to determine whether teachers’ emotional intelligence moderates the relationship between psychosocial risk factors and burnout. In a sample of teachers, we assessed psychosocial risks, emotional intelligence, and burnout and then tested whether high emotional intelligence attenuates the association between psychosocial risk factors and burnout symptoms. Although the literature suggested a potential protective role of emotional intelligence, the absence of prior studies testing this mechanism means that the present analysis is exploratory in nature, allowing for the possibility that emotional intelligence may not, in fact, moderate or mediate this relationship.

By clarifying this moderating role of emotional intelligence, the study aims to advance our understanding of how individual emotional competencies can protect against burnout in the teaching profession. Such insights have important practical implications. If emotional intelligence indeed buffers the effects of workplace stressors, then interventions that enhance teachers’ emotional intelligence (through training or professional development) alongside improvements in work conditions could be a promising dual strategy to combat teacher burnout [24].

In line with previous research emphasizing the role of personal resources in coping with occupational stress [15,23], the present study adopts an explicitly individual and psychological approach, focusing on emotional intelligence as a personal competency while acknowledging that organizational and collective factors may also influence teachers’ experiences of psychosocial risks.

## 2. Materials and Methods

### 2.1. Study Design and Ethics

In this cross-sectional study, a non-probabilistic convenience sample was collected through a snowball method among Portuguese teachers from public and private secondary schools. Participants were reached by personal network contacts who agreed to disseminate the study among secondary school teachers on social media platforms (e.g., WhatsApp and LinkedIn). Data were collected online by distributing a questionnaire via Google Forms between 6 February 2025 and 15 April 2025.

The questionnaire included various scales, starting with a cover page that briefly explained the study’s objectives. The criteria for participation involved informed consent, voluntary involvement, and confidentiality. Informed consent was obtained from all participants before they completed the survey. A total of 215 teachers completed the full questionnaire. There were no missing values in the dataset, as the questionnaire was administered via Google Forms with all items marked as mandatory. As a result, the data collected includes complete responses for every item in the study protocol. The study protocol includes 3 distinct scales, comprising a total of 91 items. Each scale is designed to assess a specific dimension: psychosocial risks, emotional intelligence, and burnout. The selection of these instruments was guided by both theoretical relevance and empirical validation, as detailed in Section 2.3. The estimated time required to complete the full questionnaire was approximately 15 min.

This study received approval from the Ethics Committee of the Faculty of Human and Social Sciences of the University of Fernando Pessoa (protocol code, Ref. FCHS/PI—475/23-4; date of approval, 20 March 2024, Porto, Portugal) and adhered to all procedures outlined in the Declaration of Helsinki.

### 2.2. Participants

The sample of this study consisted of 215 secondary school teachers of public (81.7%) and private (18.3%) schools, 73.5% of whom were female, with participant age varying from 22 to 67 years old (mean = 51.55; median = 53; SD = 9.724), the majority reported being married or in a de facto union (66.2%). A total of 74.9% held a graduate university degree, 21% a master’s degree, and 4.1% a PhD-level degree. Most of the participants were employed in public schools (81.7%) under permanent work contracts (79.5%).

### 2.3. Instruments

This study employed the Burnout Assessment Tool (BAT-23) to assess burnout dimensions, the Health and Work Survey (INSAT) to evaluate work-related psychosocial risk factors, and the Wong and Law Emotional Intelligence Scale (WLEIS-P) to evaluate emotional intelligence.

The BAT-23, developed by Schaufeli, Desart, & De Witte [28], consists of twenty-three items measuring four core symptoms of burnout: exhaustion (eight items; e.g., “At work, I feel mentally exhausted”), mental distance (five items; e.g., “I struggle to find any enthusiasm for my work”), emotional impairment (five items; e.g., “At work, I feel unable to control my emotions”), and cognitive impairment (five items; e.g., “At work, I have trouble staying focused”). The BAT-23 provides an integrated perspective, as all four dimensions are interconnected and relate to the same underlying condition. Responses to all items were recorded on a five-point Likert scale, ranging from 1 (never) to 5 (always) [28]. In this study, the Portuguese version of the Burnout Assessment Tool was used [29]. The scale establishes two cut-off points: (a) scores beginning at 2.59 indicate burnout risk, and (b) scores exceeding 3.02 suggest positive burnout diagnoses. Cronbach’s alpha values for our sample were 0.946 for all scales, 0.937 for the exhaustion core symptom subscale, 0.913 for the mental distance core symptom subscale, 0.908 for the cognitive impairment core symptom subscale, and 0.914 for the emotional impairment core symptom subscale.

This study was supported by the INSAT—Health and Work Survey, a self-reported questionnaire that measures working conditions, risk factors, and health problems [30]. Concerning the main goal of the present study, only the psychosocial risk factor scale from the INSAT was used (the PSR scale). In comparison with other psychosocial risk questionnaires, this instrument was developed based on the principle that the assessment and prevention of such risks must be grounded in work analysis. This approach entails understanding psychosocial risks within a contextualized framework, acknowledging that the selection of items was informed not only by a literature review but also by empirical studies rooted in work analysis [18,31,32,33]. The questionnaire on psychosocial risks (PSRs) comprises fifty-two items distributed across seven categories with varying numbers of items: high demands and work intensity (WI: eleven items; e.g., “Frequent interruptions”), working hours (WH: eight items; e.g., “Exceeding normal working hours”), lack of autonomy initiative (AI: four items; e.g., “Not being able to participate in decisions regarding my work”), social work relations (SWR: eight items; e.g., “Needing help from colleagues and not having it”), employment relations (ER: ten items; e.g., “I feel exploited most of the time”), emotional demands (ED: seven items; e.g., “Being exposed to the difficulties and/or suffering of other people”), and work value conflicts (WV: four items; e.g., “My professional conscience is undermined”). These categories are organized in different items. All items are measured on a 6-point Likert scale ranging from 0 (not being exposed) to 5 (being exposed with high discomfort). The Cronbach’s alpha value is 0.942 for the entire scale, and the values for each category are as follows: 0.833 for high demands and work intensity, 0.869 for working times, 0.893 for lack of autonomy, 0.931 for work relations with coworkers and managers, 0.772 for employment relations with the organization, 0.934 for emotional demands, and 0.895 for work values.

The WLEIS [34] was used to evaluate emotional intelligence. This scale is a widely used self-report instrument designed to assess emotional intelligence based on the four-branch ability model proposed by Mayer and Salovey [35]. The scale consists of sixteen items rated on a 5-point Likert scale, ranging from 1 (Strongly disagree) to 5 (Strongly agree), divided into four subscales, each containing four items: self emotion appraisal (SEA); others’ emotion appraisal (OEA); use of emotion (UOE) and regulation of emotion (ROE). In this study, we used the Portuguese version of the WLEIS [36], with strong internal consistency, with Cronbach’s alpha values ranging from 0.78 to 0.86 across subscales and high construct validity. It maintains the conceptual integrity of the original scale, and the four-factor structure was preserved with the four dimensions of emotional intelligence: (i) the evaluation and expression of emotions themselves, (ii) evaluation and recognition of emotions in others, (iii) use of emotions, (iv) regulation of emotions of one’s own. Cronbach’s alpha values for our sample were 0.859 for all scales, 0.837 for the self emotional appraisal (SEA) subscale, 0.800 for the others’ emotional appraisal (OEA) subscale, 0.785 for the use of emotion (UOE) subscale, and 0.835 for the regulation of emotion (ROE) subscale.

### 2.4. Data Analysis

Data analysis was conducted using the IBM SPSS statistical program for Windows, version 29.0 (SPSS Inc., Chicago, IL, USA) and with the PROCESS macro (model 4) for mediation analysis and the PROCESS macro (model 1) for moderation analysis [37]. A significance level of *p* ≤ 0.05 was adopted. Frequencies were used to present sociodemographic characteristics. Descriptive analysis, including range, mean, standard deviation, skewness, and kurtosis, was performed on the mean scores of psychosocial risk factor subscales, burnout subscales, and emotional intelligence subscales. Subsequently, a correlation analysis with the Pearson coefficient was performed to analyze the existing correlations. Finally, the PROCESS macro statistical tool was applied, based on the principles of ordinary least squares (OLS) linear regression, to analyze the interaction between psychosocial risks (PSRs) and burnout, with emotional intelligence (EI) as a moderator (Model 1), as well as (OLS) linear regression with bootstrapping for mediation to test several paths: (a) effect of PSRs on EI; (b) effect of EI on burnout, controlling for PSRs; (c) direct effect of PSRs on burnout, controlling for EI, and (d) indirect effect between paths a and b (model 4). The adherence to the assumptions of the method was verified, and the obtained results were deemed reliable.

Using G*Power software (Version 3.1.9.6—Mac OS X version, Heinrich Heine University Düsseldorf, Düsseldorf, Germany), a post hoc power analysis was performed to determine whether the sample size was sufficient for analysis [38]. The study showed an obtained power of 0.95, demonstrating that the sample size of 215 individuals was adequate to identify medium effects with high confidence, assuming a medium effect size (Cohen’s f^2^ = 0.10), α = 0.05, and two predictors. This result demonstrates the strength of the regression-based moderation and mediation analysis conducted with SPSS’s PROCESS macro v5.0 (Andrew F. Hayes, Calgary, AB, Canada).

## 3. Results

### 3.1. Descriptive Analysis

Table 1 presents the descriptive statistics, means and standard deviations, skewness, and kurtosis of the distributions of each subscale, namely burnout (exhaustion, mental distance, emotional impairment, and cognitive impairment); psychosocial risk factors (work intensity (WI), working hours (WH), autonomy and initiative (AI), social work relations (SWR), employment relations (ER), emotional demands (ED) and work values (WV)); and emotional intelligence (self-emotion appraisal (SEA); others’ emotion appraisal (OEA); use of emotion (UOE); and regulation of emotion (ROE)).

The data showed that all variables were approximately normally distributed, with skewness indices (|γ_1_|) 0 and kurtosis indices (|γ_2_|) within ±2, meeting the recommended thresholds for normality [39].

### 3.2. Correlation Analysis

A correlation analysis is a prerequisite to support a mediation/moderation analysis. To ensure statistical validity, correlation analysis was performed using the Pearson coefficient applied between (1) PSR subscales and BAT-23 subscales to show whether psychosocial risk factors are associated with burnout (Table 2); (2) WLEIS subscales and BAT-23 subscales to show whether emotional intelligence may reduce burnout (justifies path b or moderating role) (Table 3); and (3) PSR subscales and WLEIS subscales to show whether emotional intelligence might be impacted by psychosocial risk factors (justifies path “a” as a mediator) (Table 4).

The data shows statistically significant, moderate-to-strong positive correlations between the psychosocial risk factors subdimensions and the burnout subdimensions, based on Cohen’s criteria [40].

Data shows statistically non-significant correlations, indicating a weak effect between the emotional intelligence subdimensions and the burnout subdimensions [40].

The data shows some weak statistically significant correlations between emotional intelligence and psychosocial risks, particularly in emotional demands (ED) and working hours (WH) with the ROE subdimension and in autonomy and initiative (AI) and social work relations (SWR) with the OEA subdimension. All other correlations are weak and not statistically significant. This means that ROE (regulation of emotion) is the most consistent protective factor, significantly negatively correlated with WH and ED. This implies that teachers who are better at managing their emotions feel less emotionally burdened by job demands. The data also shows that OEA negatively correlates with the autonomy and social relationship subscales. This may suggest that those who are more emotionally attuned to others are also more sensitive to interpersonal stressors, or more skilled in navigating them.

### 3.3. Mediation and Moderation Analysis

A mediation analysis using the PROCESS macro (Model 4) was conducted to examine whether emotional intelligence mediates the relationship between psychosocial risk factors and burnout among teachers. The results are presented in Table 5.

The data shows that the total effect of psychosocial risk factors on burnout was significant, indicating that higher perceived risk predicts greater burnout (B = 0.312, SE = 0.021, *p* < 0.001). The effect of psychosocial risk on emotional intelligence was not significant (B = −0.034, SE = 0.018, *p* = 0.057), suggesting that emotional intelligence is not significantly affected by stress levels. Emotional intelligence, when included in the model, was a significant positive predictor of burnout (B = 0.174, SE = 0.080, *p* = 0.031), indicating a suppressor effect, as this direction contradicts theoretical expectations. The indirect effect of psychosocial risk on burnout through emotional intelligence was not significant (B = −0.006, 95% CI [−0.0169, 0.0011]). Figure 1 helps to better visualize the mediating role of emotional intelligence in the relationship between psychosocial risks and burnout.

These findings do not support a mediating role of emotional intelligence in the relationship between psychosocial risk and burnout.

A moderation analysis was performed using the PROCESS macro (Model 1) to test whether emotional intelligence moderates the effect of psychosocial risk on burnout. The overall model was significant and explained 51% of the variance in burnout, *R*^2^ = 0.508, *F* (3, 211) = 72.65, *p* < 0.001. The results are presented in Table 6.

The overall model was significant and explained 51% of the variance in burnout, *R*^2^ = 0.508, *F* (3, 211) = 72.65, *p* < 0.001.

Psychosocial risk was a significant positive predictor of burnout (*B* = 0.313, *p* = 0.001), indicating that higher perceived risk was associated with higher burnout symptoms. Emotional intelligence did not significantly predict burnout on its own (*B* = 0.176, *p* = 0.364), and the interaction term (PSR × EI) was not significant (*B* = 0.000, *p* = 0.995), indicating that emotional intelligence does not moderate the relationship between psychosocial risks and burnout.

Figure 2 helps to better visualize the mediation role of emotional intelligence on the relation between psychosocial risks and burnout.

## 4. Discussion

The purpose of this paper was to determine whether teachers’ emotional intelligence moderates the relationship between psychosocial risk factors and burnout.

A central concern emerging from this research is the intensity of burnout symptoms, with particular emphasis on the pronounced emotional exhaustion experienced by teachers. As expected, and in line with several recent studies [2,41,42], the data shows alarmingly high levels of the exhaustion dimension of burnout in the teaching profession. In fact, teachers are considered the most vulnerable workers, susceptible to burnout and emotional exhaustion—characterized by chronic fatigue, depleted emotional resources, emotional fragility, and a sense of being overwhelmed—which is recognized as the most debilitating dimension of burnout in the teaching profession [43,44].

This study emphasizes elevated levels of psychosocial risks (PSRs) that can be associated with increased complaints of emotional exhaustion. The correlations confirmed that psychosocial risk factors, particularly emotional demands, working hours, lack of autonomy and initiative, and work values, were strongly and positively associated with burnout symptoms such as exhaustion and were moderately and positively associated with mental distance and emotional impairment. These findings are aligned with existing research highlighting the central role of job stressors in the development of teacher burnout [7,25,41,45,46].

Data shows that, overwhelmingly, emotional demands, working hours, and a lack of autonomy and working values are the main psychosocial risk factors with the strongest association with elevated levels of teacher burnout, particularly in the dimension of emotional exhaustion. These factors reflect the cumulative effect of these demands, which induce chronic strain, conflicting demands, and emotionally charged interactions, which contributes to a sustained depletion of emotional resources, reinforcing the vulnerability of educators to burnout.

In fact, psychosocial risk factors have a significant influence on the manifestation of burnout symptoms, highlighting the importance of research in this field for understanding the widespread impact of these risks on teachers. Data shows that teachers are exposed to high emotional demands (managing diverse classroom behaviors, dealing with students’ difficulties and fears, and maintaining discipline), which can be stressful and anxiety-inducing. This is reinforced in the results obtained in this study; high job demands and low social support—two prominent psychosocial risk factors— directly contribute to emotional exhaustion and psychological strain, positioning teaching among the most stress-vulnerable professions [2,47,48]. When combined with other psychosocial risks found in our study, like working hours (extensive workload, including lesson planning, grading, and extracurricular activities) and a lack of autonomy and initiative, this can contribute to chronic stress, burnout, and a decline in overall mental well-being and increase the prevalence of burnout symptoms.

Educators face a complex web of psychosocial risks that significantly affect their physical, emotional, and social well-being [49,50,51,52], indicating that burnout is not merely a localized phenomenon but a systemic issue exacerbated by increasing workloads, emotional labor, and insufficient institutional support.

Several studies indicate that teachers’ emotional intelligence is a key predictor of psychological well-being and a safeguard against burnout [53,54]; this is reinforced by recent systematic reviews [15,23] that specifically analyze the relationship between emotional intelligence and teacher burnout, offering strong evidence that emotional intelligence acts as a protective factor. Moreover, emotional skills used to identify feelings and emotions facilitate more effective emotional strategies to deal with negative events; teachers with more emotional competencies are better prepared to handle strain and emotional burden and develop regulation strategies to handle emotional exhaustion [21,55].

However, in our study, the mediation model shows new insights into the role of emotional intelligence (EI) through psychosocial risk factors (PSRs) and burnout. Our results suggest that the protective role of emotional intelligence is not enough to provide teachers with lower levels of burnout.

If we assume that emotional intelligence is the capacity for self-awareness, as well as the ability to identify the emotions, feelings, and needs of others, with a view to establishing cooperative relationships and to achieve effective problem-solving and decision-making [20,56,57], our results show that emotional intelligence can be helpful but is not sufficient in preventing burnout. In fact, while emotional intelligence has been widely recognized as a protective factor against burnout, the results suggest that its role in preventing burnout may be more limited than previously assumed.

The analysis of emotional intelligence data showed weak and inconsistent correlations with both psychosocial risks and burnout. Although there were two dimensions (such as others’ emotion appraisal (OEA) and regulation of emotion (ROE)) that showed a modest negative correlation with emotional demands and emotional impairment, the extent and strength of these effects were constrained.

The theoretical role of emotional intelligence as a psychological resource, both in the mediation and moderation models, was analyzed using the PROCESS macro [37]. The mediation analysis (Model 4) showed no significant indirect effect of psychosocial risk on burnout through emotional intelligence. Despite the regression model’s weak association between emotional intelligence and burnout, the impact’s unexpectedly positive direction raises the possibility of a measurement overlap or suppression effect.

Results from the moderation analysis (Model 1) were likewise not statistically significant. Effect of psychosocial risk on burnout was not mitigated by emotional intelligence, and the interaction term (PSR × EI) added no new explanatory power to the model.

Overall, these results suggest that emotional intelligence does not play a strong role in reducing or buffering burnout, whereas psychosocial risk factors remain strong and consistent predictors of burnout. This prompts an important reflection on whether emotional intelligence, on its own, is sufficient to mitigate the adverse effects of occupational stressors among teachers, particularly when exposure to high levels of psychosocial risks may override or diminish its protective role.

Our findings are consistent with the previous literature [58]: although EI was inversely related to burnout, the strength of this relationship was weak to moderate, indicating that EI alone may not be sufficient to buffer individuals—particularly educators and healthcare professionals—from chronic stress and emotional fatigue. Souza and Lima [59] highlighted that not all components of EI contribute equally to burnout prevention, and that contextual and organizational factors often outweigh individual emotional competencies. Furthermore, while emotional intelligence facilitates emotional regulation, it needs to be complemented by adaptive coping strategies and systemic interventions to effectively mitigate burnout [57].

It is important to note that, while this study focuses on individual-level protective factors, particularly teachers’ emotional intelligence, in mitigating burnout, protective organizational resources (e.g., positive school climate, teacher autonomy, and institutional support) are also crucial. Individual traits such as emotional intelligence do not operate in isolation and their protective value is contingent on the broader organizational context. For instance, supportive school environments and adequate institutional resources significantly reduce burnout risk, and burnout often arises from mismatches between individual capacities and the organizational environment. Indeed, burnout is not solely an individual condition but also a phenomenon shaped by the workgroup and organizational context. Thus, even as we focus on personal resources, we acknowledge that a supportive organizational context is essential for enabling and amplifying the protective effects of individual factors [45]. As emphasized by Philippe Askenazy [60], the organization and intensity of work itself shape workers’ health and capacity to mobilize protective individual factors.

Moreover, while our study focused primarily on the role of emotional intelligence as an individual protective factor, we acknowledge that additional individual characteristics, such as age, gender, and professional seniority, as well as the collective organization of the teaching profession, are likely to shape both the experience of psychosocial risks and the development of protective mechanisms. Recent conceptual work on psychosocial risks emphasizes that burnout emerges from the complex interplay between individual, organizational, and collective dimensions [61,62]. Our findings should therefore be interpreted with caution, as the omission of these contextual and situational variables limits the scope of our conclusions. Future studies should adopt a more comprehensive analytical framework to investigate how personal characteristics and collective work dynamics interact with organizational factors to influence burnout risk among teachers.

These findings underscore the need for a more holistic approach to addressing burnout—one that places teachers’ work activity at the center of intervention strategies—improving working conditions and organizational support. Rather than relying solely on individual emotional competencies, such as emotional intelligence, this perspective includes integrating organizational context determinants (time pressure, classroom disruption, workload stressors, technical and administrative difficulties, disruptive class management, and work climate).

A teacher-centered approach to burnout prevention is essential to move beyond individual psychological resources and to address organizational dimensions of the profession itself. Reshaping the conditions under which teachers carry out their professional responsibilities—alongside prioritizing collaborative practices, professional autonomy, and the recognition of effort and expertise—can significantly enhance emotional and psychological well-being. Such a systemic approach plays a pivotal role in reducing burnout in a sustainable and effective way and is of paramount importance for the educational system’s capacity to create conditions that value and sustain teachers, following the idea of a holistic health and well-being.

This study is subject to some limitations that should be acknowledged when interpreting the findings. The sample presents a notable gender imbalance, with a predominance of female participants, which may limit the generalizability of the results across the teaching population. The age distribution of the participants is imbalanced; although the range spans from 22 to 67 years old, most participants aged around 50 years (mean = 51.55; median = 53; SD = 9.724), restricting the possibility of a meaningful analysis of age differences. The representation of school types was disproportionate, with a significantly higher number of teachers from public schools (over 80%), which may have influenced the observed relationships. As such, the obtained results cannot be solidly generalized to the population, and group comparisons are hampered. Another limitation may be related to school organizational factors, which could perhaps play a part in increasing or decreasing burnout among teachers, variables that could not be included in this study but that can significantly impact the validity of the results and their interpretation.

Future research should analyze the effects of school organizational climate and leadership, explore potential age and gender differences, and account for the hierarchical nesting of teachers within schools (public and private) or regions. This cross-sectional design with a self-reported questionnaire may be expanded to enlarge the sample and adopt stratified or probability-based sampling methods to enhance generalizability. Replicating this study in different populations would help confirm the validity of the findings.

## 5. Conclusions

Our findings provide relevant insights that may have useful theoretical and practical implications. From a theoretical perspective, further research is needed to deepen the analysis of psychosocial risk differences. It is important to confirm whether the results of the present study show that some psychosocial risk factors can have different impacts on burnout. Teaching is characterized by a complex set of psychosocial demands, including high work intensity, extended working hours, and elevated emotional demands. These factors may interact in ways that amplify their effects on teacher well-being. Indeed, teachers face a wide range of adverse social and psychological consequences as a result of burnout. Those psychological effects have been widely reported in the scientific literature and include emotional exhaustion, reduced professional efficacy, strained interpersonal relationships, and a diminished sense of personal accomplishment. Emotional intelligence, particularly emotional regulation, plays an important role in coping with adversity; however, it is not sufficient on its own to prevent or mitigate burnout. Burnout is a multifaceted phenomenon deeply rooted in organizational, social, and working conditions which demands comprehensive and sustained interventions.

## Figures and Tables

**Figure 1 ijerph-22-01439-f001:**
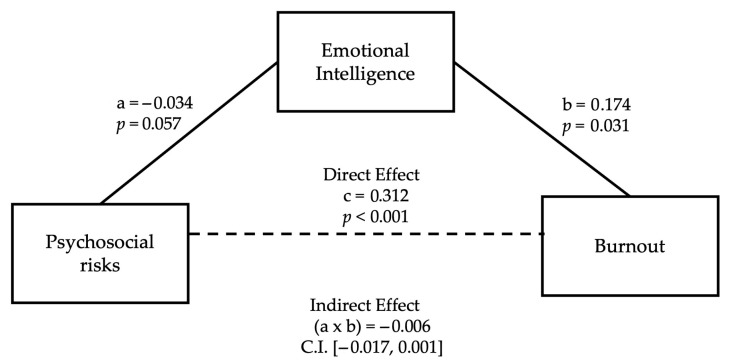
Mediation model: EI as mediator between PSR and burnout.

**Figure 2 ijerph-22-01439-f002:**
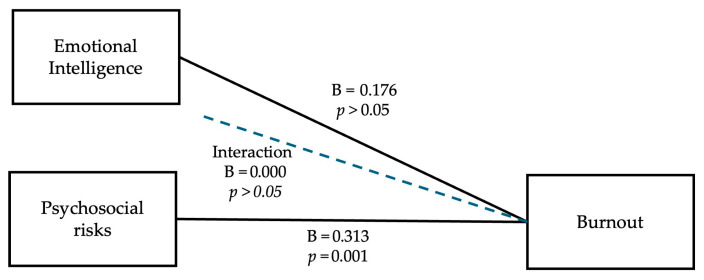
Moderation model: EI as moderator between PSR and burnout.

**Table 1 ijerph-22-01439-t001:** Descriptive analyses for PSR, BAT-23, and WLEIS subscales.

Scale Subscale	Min	Max	Mean	SD	Skewness		Kurtosis	
					γ_1_	SE	γ_2_	SE
PSR								
WI	0.00	4.09	2.2567	0.77357	−0.314	0.166	0.067	0.330
WH	0.00	3.75	1.9302	0.91462	0.055	0.166	−0.899	0.330
AI	0.00	5.00	1.9372	1.45297	0.294	0.166	−1.187	0.330
SWR	0.00	3.88	0.9186	0.99079	1.182	0.166	0.714	0.330
ER	0.00	3.90	1.8107	0.81437	−0.123	0.166	−0.062	0.330
ED	0.00	4.38	2.0145	1.25891	0.323	0.166	−0.986	0.330
WV	0.00	3.75	1.3884	1.09041	0.515	0.166	−0.708	0.330
BAT-23								
Exhaustion	1.13	5.63	3.3860	1.01742	0.368	0.166	−0.272	0.330
Mental Distance	0.80	4.00	1.4977	0.74102	1.215	0.166	0.932	0.330
Cognitive Impairment	1.00	5.00	2.2251	0.88472	0.708	0.166	0.268	0.330
Emotional_Impairment	1.00	5.00	2.1070	0.89563	1.026	0.166	1.051	0.330
WLEIS								
SEA	1.00	5.00	2.9709	1.06397	−0.748	0.166	−0.659	0.330
OEA	1.00	5.00	3.0965	0.95625	−0.858	0.166	−0.136	0.330
UOE	0.75	3.75	2.2593	0.77748	−0.574	0.166	−0.483	0.330
ROE	0.75	3.75	2.3535	0.71658	−0.584	0.166	−0.059	0.330

Legend: SD—standard deviation; SE—standard error; γ_2_ — gamma squared.

**Table 2 ijerph-22-01439-t002:** Pearson correlations between PSR subscales and BAT-23 subscales.

	PSR						
BAT-23	WI	WH	AI	SWR	ER	ED	WV
Exhaustion	0.480	0.647	0.613	0.508	0.504	0.656	0.645
Mental Distance	0.407	0.457	0.504	0.534	0.378	0.536	0.552
Cognitive Impairment	0.354	0.415	0.355	0.436	0.320	0.445	0.540
Emotional Impairment	0.377	0.430	0.396	0.503	0.392	0.512	0.537

NOTE: All correlations are significant at the 0.01 level (2-tailed).

**Table 3 ijerph-22-01439-t003:** Pearson correlations between psychosocial risk factor subscales and burnout subscales.

	WLEIS			
BAT-23	SEA	OEA	UOE	ROE
Exhaustion	0.029	−0.105	0.006	−0.100
Mental Distance	0.081	0.025	−0.004	−0.071
Cognitive Impairment	0.066	0.103	0.074	−0.028
Emotional Impairment	0.035	0.013	0.050	−0.087

NOTE: All correlations are non-significant at the 0.05 level (2-tailed).

**Table 4 ijerph-22-01439-t004:** Pearson correlations between psychosocial risk subscales and emotional intelligence subscales.

	PSR						
WLEIS	WI	WH	AI	SWR	ER	ED	WV
SEA	−0.020	0.060	0.006	−0.006	−0.043	−0.032	−0.078
OEA	−0.118	−0.101	−0.134 *	−0.138 *	−0.070	−0.129	−0.067
UOE	−0.064	−0.015	−0.044	0.010	−0.074	−0.115	−0.067
ROE	−0.132	−0.160 *	−0.122	−0.073	−0.122	−0.180 **	−0.116

Note: ** Correlation is significant at the 0.01 level and * correlation is significant at the 0.05 level (2-tailed).

**Table 5 ijerph-22-01439-t005:** Mediation analysis.

Path	B	SE	t	*p*	95% CI
a (PSR → EI)	−0.034	0.018	−1.91	0.057	[−0.070, 0.001]
b (EI → Burnout)	0.174	0.080	2.18	0.031	[0.016, 0.332]
c (Direct effect)	0.312	0.021	14.80	<0.001	[0.271, 0.354]
Indirect effect	−0.006	0.005	–	–	[−0.017, 0.001]

Legend: CI—confidence interval.

**Table 6 ijerph-22-01439-t006:** Moderation analysis.

Predictor	B	SE	t	*p*	95% CI
Constant	17.94	9.72	1.85	0.066	[−1.23, 37.10]
PSR	0.313	0.090	3.47	0.001	[0.14, 0.49]
WLEIS	0.176	0.193	0.91	0.364	[−0.21, 0.56]
PSR × WLEIS	0.000	0.002	−0.01	0.995	[−0.004, 0.004]

Legend: CI—confidence interval.

## Data Availability

The data presented in this study are available on request from the corresponding author.

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
