# Peer review of "Psychosocial Risk Factors and Burnout Among Teachers: Can Emotional Intelligence Make a Difference?"

_ijerph, 2025, doi:10.3390/ijerph22091439_

Round 1
Reviewer 1 Report
Comments and Suggestions for Authors
The manuscript is well structured and basically has a good research idea. On the other hand, there are elements that cause concern.
1. What is the purpose of research? At the end of the Introduction section, the objective of the research is clearly defined. Contrary to expectations, the results of the study showed that emotional intelligence plays neither a mediating nor a moderating role. If it is questionable whether emotional intelligence has a mediating/moderating role, then in the review of the literature in the Introduction section it is necessary to provide an overview of previous research that argues for such a position. The existing literature review predicts the expectation to confirm the mediating/moderating role of emotional intelligence, not leaving the possibility for the opposite outcome - that it is not confirmed.
2. It is necessary to pay more attention to the part related to sampling. How is the sample size determined? Is the sample representative? A clear argumentation is required.
3. Statistical analysis has significant shortcomings that need to be eliminated. The assumption for the application of SEM analysis is factor analysis. In the section related to the results, the results of the factor analysis were completely absent. It is necessary to present measurement items, factor loadings, AVE, CR and Cronbach's Alpha for all latent variables. Also, it is necessary to test discriminant validity through the Fornell-Larcker criterion, as well as model fit statistics.
This is a critical aspect of the study. The lack of confirmation of the mediating/moderating role of emotional intelligence may also be a consequence of poorly conducted research, rather than the factual situation as such. In order to eliminate causes related to the design of the study and the procedure of sampling and research, it is necessary to conduct the mentioned analyzes and argue the validity of the study.
4. The terms "must" and "crucial" are used several times in the text. This is not a good academic practice and therefore better alternatives need to be found.
5. What are the limitations of the study? In a fair and consistent way, it is necessary to state the limitations of the study.
Author Response
Dear Reviewer,
Thank you very much for taking the time to review this manuscript. Please find the detailed responses below and the corresponding revisions/corrections highlighted/in track changes in the re-submitted files.

Reviewer 2 Report
Comments and Suggestions for Authors
Dear authors, thank you for the opportunity to get acquainted with the results of your study.
The study is devoted to establishing the influence of emotional intelligence on the relationship between psychosocial factors and teacher burnout. The authors convincingly show that the emotional intelligence of teachers is not enough to reduce burnout under the influence of psychosocial factors in the workplace.
In the introduction, the authors analyzed scientific research and approaches on the topic, substantiated the goal and hypothesis of the study.
The study sample is representative, the methods are reliable, the statistical procedures are correct and correspond to the goal of the study. The authors presented the results in a very structured manner. There are many tables and explanations for them. Not only the main hypothesis of the study was tested, but also the key psychosocial factors at work associated with teacher burnout were identified.
The authors conducted a detailed discussion of the results, compared the results of the study with other authors, and indicated possible reasons why the hypothesis was not confirmed. The limitations of the study are indicated.
There are a number of recommendations and questions regarding the text of the manuscript:
1. Did the authors study the issue of the influence of age and experience on burnout? and also on the studied relationships between psychosocial factors and emotional intelligence? Could this factor have influenced the result?
2. Have differences in psychosocial factors and their impact on burnout been studied depending on the place of work (private or public, or various public organizations)?
3. In the discussion of the results, it is possible to add what other mitigating factors of work and personal factors of teachers can contribute to reducing burnout. What could be clarified in future studies.
4. I would like to specify the conclusions under the main hypothesis, as well as supplement the information on which psychosocial factors are more associated with the development of teacher burnout.
The manuscript requires minimal revision before publication.
Best wishes, reviewer
Author Response

(The authors gave the same response as above.)

Reviewer 3 Report
Comments and Suggestions for Authors
Dear Authors,
Congratulations on your work. I was pleased to read your manuscript and hope that the comments below will assist in strengthening it.
1. In the Introduction section, the authors justify the need to address protective factors (on p. 2; lines 68-70). A few lines down, it is stated that “emotional intelligence has emerged as a potentially important protective factor in the teaching profession”. The way the authors address possible protective factors gives the impression that we're talking mainly about individual factors. Is this idea correct? If so, in my opinion, the authors are disregarding important protective factors that are built collectively and as a function of the organisational resources available. For instance, the expression of individual protective factors is always conditioned by the leeway permitted by the organisational context. In other words, this is not a direct relationship, unaffected by the working environment. In this regard, the authors may see Yves Clot’s or Philippe Askenazy’s works.
On the other hand, the authors also fail to consider individual variables that are important for understanding the construction of protective factors, such as age, gender, seniority and the preponderant role of work collectives, or the collective organisation of the profession. For an overview on this subject, the authors can consult: https://www.etui.org/publications/conceptualising-work-related-psychosocial-risks; or https://travail-emploi.gouv.fr/sites/travail-emploi/files/files-spip/pdf/rapport_SRPST_definitif_rectifie_11_05_10.pdf
With these remarks, I do not intend to question the validity of the research underpinning the authors’ hypotheses. However, I believe that the study overlooks important contextual and situational variables that are essential for understanding the relationship between psychosocial risk factors and their health-related consequences (burnout, in this case).
Therefore, the authors should either explicitly state that their approach is rooted in an “individual and psychological approach” (in line with transactional models), or they should consider addressing the nature of psychosocial risk factors in greater depth.
That said, on page 11 (Discussion section), the authors noted: “These findings underscore the need for a more holistic approach to addressing burnout – one that places teachers’ work activity at the centre of intervention strategies – improving working conditions and organisational support. Rather than relying solely on individual emotional competencies, such as emotional intelligence, this perspective includes integrating organisational context determinants”. This conclusion, however, is well-established in the literature, which has long demonstrated the limitations of purely individual and psychological approaches to assessing PSR. In other words, this recognition should not appear solely in the discussion as if it were a novel insight of this study. It must be acknowledged and integrated into the theoretical framework from the outset. For this, it is recommended, for example, to consider works in the field of activity-oriented approaches to work psychology and ergonomics.
2. Regarding the instruments employed, the authors should provide a justification for choosing the INSAT questionnaire over other instruments designed to assess PSR. What are the strengths of INSAT compared to other available tools? Why was it considered the most appropriate for this study?
In addition, there appears to be some inconsistency in how the INSAT questionnaire is described. On page 2, it is introduced as a tool to assess working conditions, whereas on page 4, it is referred to as “the questionnaire on psychosocial risks (…)”. This requires clarification.
Furthermore, it would be important to specify the scope of the INSAT as used in this study. How many items does the INSAT contain in total? Was the full questionnaire administered, or were only those items related to PSR included in the analysis?
3. In the Method section (viz., Study design and ethics), the authors stated that "The estimated time to complete the questionnaire was about 15 minutes". However, the total number of items included in the questionnaire made available to teachers is not mentioned. This information would be important to assess the feasibility and response burden for participants.
4. In the Participants section, it is indicated that the sample comprises 215 teachers. However, no information is provided regarding the percentage of missing values. As it stands, this wording suggests that all 215 teachers completed the full questionnaire, which seems unlikely. Conversely, if the 215 represents the final sample after excluding responses with a certain level of missing data, then the authors should clearly state how many questionnaires were initially received and how many were excluded due to incomplete responses.
5. At the end of page 8 and the beginning of page 9, the same sentence appears to be repeated. Additionally, the manuscript would benefit from a thorough revision of formatting according to MDPI’s submission guidelines.
6. Regarding the Discussion section, some of my concerns have already been outlined in comment 1. Nevertheless, it would be valuable for the authors to provide more concrete suggestions. For example, when referring to a “teacher-centred approach”, could the authors elaborate on what specific intervention measures they are referring to? How might these be designed and implemented in practice? What role should teachers themselves play in these processes? Again, several works in the field of work psychology could provide key clues to address this issue.
Moreover, when the authors suggest that “(…) it might be important to analyse the role of school organizational climate and leadership and, also, age and gender differences”, but it would be equally relevant to consider the type of school – public or private. As highlighted in the literature, working conditions can differ substantially between these two contexts, and such differences may have implications for how PSR manifest and are experienced by teachers.
7. The final list of references should be revised to ensure full compliance with the journal’s referencing style guidelines. This includes, for example, the use of italics, punctuation, and formatting of DOIs, volume or issue numbers.
Once again, congratulations on your work. I hope my comments help you somehow improve your manuscript.
Best.
Author Response

(The authors gave the same response as above.)

Round 2
Reviewer 1 Report
Comments and Suggestions for Authors
No additional comments.
Reviewer 3 Report
Comments and Suggestions for Authors
Dear authors,
Congratulations on your work. The revised version of your manuscript fully addresses all my comments.
All the best.